# Neuroprotection of Retinal Ganglion Cells with AAV2-BDNF Pretreatment Restoring Normal TrkB Receptor Protein Levels in Glaucoma

**DOI:** 10.3390/ijms21176262

**Published:** 2020-08-29

**Authors:** Anna Wójcik-Gryciuk, Olga Gajewska-Woźniak, Katarzyna Kordecka, Paweł M. Boguszewski, Wioletta Waleszczyk, Małgorzata Skup

**Affiliations:** 1Laboratory of Neurobiology of Vision, Nencki Institute of Experimental Biology, 02-093 Warsaw, Poland; a.wojcik-gryciuk@nencki.edu.pl (A.W.-G.); k.kordecka@nencki.edu.pl (K.K.); w.waleszczyk@nencki.edu.pl (W.W.); 2Mediq Clinic, 05-120 Legionowo, Poland; 3Group of Restorative Neurobiology, Nencki Institute of Experimental Biology, 02-093 Warsaw, Poland; o.gajewska-wozniak@nencki.edu.pl; 4Laboratory of Behavioral Methods, Nencki Institute of Experimental Biology, 02-093 Warsaw, Poland; p.boguszewski@nencki.edu.pl

**Keywords:** retina, BDNF overexpression, TrkB receptor downregulation, trabecular occlusion, microbeads

## Abstract

Intravitreal delivery of brain-derived neurotrophic factor (BDNF) by injection of recombinant protein or by gene therapy can alleviate retinal ganglion cell (RGC) loss after optic nerve injury (ONI) or laser-induced ocular hypertension (OHT). In models of glaucoma, BDNF therapy can delay or halt RGCs loss, but this protection is time-limited. The decreased efficacy of BDNF supplementation has been in part attributed to BDNF TrkB receptor downregulation. However, whether BDNF overexpression causes TrkB downregulation, impairing long-term BDNF signaling in the retina, has not been conclusively proven. After ONI or OHT, when increased retinal BDNF was detected, a concomitant increase, no change or a decrease in TrkB was reported. We examined quantitatively the retinal concentrations of the TrkB protein in relation to BDNF, in a course of adeno-associated viral vector gene therapy (AAV2-BDNF), using a microbead trabecular occlusion model of glaucoma. We show that unilateral glaucoma, with intraocular pressure ( IOP) increased for five weeks, leads to a bilateral decrease of BDNF in the retina at six weeks, accompanied by up to four-fold TrkB upregulation, while a moderate BDNF overexpression in a glaucomatous eye triggers changes that restore normal TrkB concentrations, driving signaling towards long-term RGCs neuroprotection. We conclude that for glaucoma therapy, the careful selection of the appropriate BDNF concentration is the main factor securing the long-term responsiveness of RGCs and the maintenance of normal TrkB levels.

## 1. Introduction

Increased intraocular pressure (IOP) remains a major, though not the only, risk factor for developing glaucoma. Available therapies that reduce IOP do not always prevent the progressive death of retinal ganglion cells (RGCs) and the development of a pathology that leads to blindness. There is therefore a need to develop an effective neuroprotection method that can interact with cellular signaling and that promotes RGCs survival. The vast majority of RGCs contain both brain-derived neurotrophic factor (BDNF) and its high-affinity TrkB receptor, raising the possibility that locally produced BDNF may play an important role in the activation of RGC through TrkB receptors in addition to the retrogradely transported ligand [1,2]. Exogenous BDNF retinal supplementation used in early experimental trials has shown promising results, while indicating that the duration of BDNF activity is limited and requires repeated administration to obtain long-term neuroprotection [3,4,5,6,7,8,9,10]. Intraocular injections of adeno-associated viral vectors carrying BDNF gene construct (AAV-BDNF) are an attractive alternative for eliciting a long-term increase in BDNF concentration in the retina and supporting the survival of RGCs [11,12,13,14,15]. This approach has also been shown to be superior to BDNF treatment directed to the neurotrophin-releasing superior colliculus, which is targeted by the RGCs. Viral vector-induced BDNF overexpression in the superior colliculus, thought to boost the retrograde neurotrophin delivery, did not increase BDNF in the retina and failed to protect RGCs in the glaucoma models used [16].

To secure therapeutic action, an effective intraocular concentration of BDNF requires that the expression of the BDNF high-affinity TrkB full-length receptor in retinal cells is maintained. Increased IOP was reported to change BDNF and TrkB availability owing to the impairment of retrograde transport in the optic nerve [17,18,19]. Furthermore, in several in vitro and in vivo systems, a high BDNF concentration was shown to downregulate its TrkB receptors, possibly affecting BDNF signaling [20,21,22,23,24,25]. However, whether intraocular BDNF overexpression leads to TrkB downregulation, impairing BDNF signaling in the retina, has not been conclusively proven [16,25]. Numerous studies show that after optic nerve injury and ocular hypertension, when increased BDNF mRNA and protein in the retina are detected, concomitant increase [16,17], no change [26] or a decrease in TrkB [27] occur, both early and long-term post-injury. Answering the question concerning the dependence of TrkB expression on BDNF changes in the glaucoma pathology and after treatment is crucial for designing efficient clinical trials. Therefore, we undertook this study to validate the neuroprotective efficacy of the AAV2-BDNF transgene, measuring the concentrations of BDNF and TrkB proteins in the retina of AAV2-BDNF-treated and untreated rats and comparing them to naive controls. Our assumption was that if overexpressed BDNF shows significant neuroprotective effects, its cognate receptor TrkB protein expression is not suppressed.

We have chosen a microbead trabecular occlusion model of glaucoma, with rapid elevation of IOP, which serves as the equivalent of a severe glaucoma attack [28]. Contralateral fellow eyes served as untreated controls—a common approach used to eliminate the factor of individual variability [16,28,29].

We will show that unilateral glaucoma leads to the bilateral reduction in BDNF concentration, accompanied by up to a four-fold TrkB upregulation. An increase in TrkB concentration did not compensate the reduced BDNF signals sufficiently to promote signaling towards the protection of the retina from significant cell loss after six weeks of pathology. Significantly, we found out that long-term, moderate BDNF overexpression in glaucomatous eye restores normal TrkB levels. In effect, increased BDNF signaling led to long-term RGCs neuroprotection both in the central and peripheral regions of the retina. We conclude that in severe glaucoma attacks, (1) a single administration of AAV2-BDNF is sufficient for significant RGCs neuroprotection, and (2) for glaucoma therapy, the careful control of BDNF concentration is the main factor securing the long-term responsiveness of RGCs and the maintenance of TrkB normal levels.

## 2. Results

### 2.1. Microbead Trabecular Occlusion Model of Glaucoma Causes an Increase in IOP both Immediately and Lasting a Few Weeks

A modification of the rat experimental model of glaucoma, incorporating rapid-onset elevation of intraocular pressure (IOP) [30], has been reproduced successfully. IOP values for untreated Left eyes and microbead-injected Right eyes are shown in Figure 1.

IOP values significantly increased immediately after microbead injection. A rapid, over five-fold elevation of IOP with an initial “high-pressure injury” confirms that the model may be used as the equivalent of a severe glaucoma attack [30]. During the first week, the IOP remained elevated four-fold, and gradually decreased between the 2nd and 5th weeks to reach close to normal values at 6 weeks. The IOP was comparable in the glaucomatous eyes in the Glaucoma and AAV-BDNF + Glaucoma groups. This result indicates that BDNF elevation owing to AAV2-BDNF administration (see below), which preceded glaucoma induction by 3 weeks, did not influence the mechanisms related to aqueous humor circulation.

### 2.2. Elevated IOP in Microbead Trabecular Occlusion Model of Glaucoma Leads to Significant RGCs Loss in Central and Peripheral Retina

The effect of glaucoma and AAV-BDNF transgene administration on the density of retinal ganglion cells (RGCs) was examined by comparing glaucomatous and non-treated, contralateral retinas (see Figure 2). We compared these effects to the RGC numbers in the retinas from the intact rats (Figure 3).

In naive animals, the number of RCGs per mm^2^ amounted to 1686 ± 30 in the central retina, and to 1250 ± 133 in the peripheral zone (mean ± SD from three rats; five retinas).

After 6 weeks, the rats with glaucoma induction showed a 31% loss of RGCs in the central retina of the glaucomatous eye, as compared to the fellow eye (two-tailed ANOVA test with multiple comparisons, post-hoc Sidak test, *F* (1,13) = 172.6, *p* < 0.001; Figure 3A, blue). The number of RGCs per mm^2^ in the glaucomatous retina amounted to 1071 ± 137, while in the contralateral retina it was 1556 ± 145.

In rats which received BDNF transgene, the loss of RGCs in the glaucomatous retina (*p* < 0.001) was markedly reduced (to approx. 15%) as compared to rats with no BDNF gene transfer (two-tailed ANOVA test with multiple comparisons, post-hoc Sidak test, *p* = 0.0038; Figure 3B, orange). In the AAV2-BDNF-treated group, the density of RGCs in the glaucomatous retina amounted to 1267 ± 60, while in the contralateral retina it was 1500 ± 49. The importance of BDNF overexpression for the protection of RGCs from degeneration in the central part of the glaucomatous retina has been confirmed by the significant interaction of glaucoma with BDNF overexpression (*p* = 0.0005).

As shown on Figure 3B, in the peripheral zone of the retina there was a 29% loss of RGCs (glaucomatous vs. contralateral eye; two-tailed ANOVA test with multiple comparisons, post-hoc Sidak test, *F* (1,12) = 67.9, *p* < 0.001). The number of RGCs per mm^2^ in the glaucomatous retina amounted to 758 ± 124, while in the contralateral retina it was 1060 ± 124. In the glaucomatous retina of the rats which received the BDNF transgene, more RGCs survived than in the untreated rats (*p* = 0.038 vs. untreated eye; two-tailed ANOVA test with multiple comparisons, post-hoc Sidak test). In that group, the density of RGCs in the glaucomatous retina amounted to 912 ± 37 and differed by 7% only from the non-treated retina (982 ± 82). Again, the importance of BDNF overexpression for the protection of RGCs in the peripheral zone has been confirmed by the significant interaction of glaucoma with BDNF overexpression (*p* = 0.0003).

### 2.3. Retinal AAV-BDNF Injection Raises BDNF beyond Normal Concentration and Restores to Normal the Upregulated TrkB Levels in the Rats with Induced Glaucoma

Next, we performed ELISA to answer the question of whether at six weeks after the induction of glaucoma BDNF protein concentrations in the retina are reduced, and if so, whether a deficit also develops in the contralateral eye. To examine whether AAV2-BDNF pretreatment effectively upregulates BDNF and whether the effect is unilateral, measurements were done in rats without and with intraocular administration of the AAV2 vector carrying the BDNF transgene to the Right eye. In parallel, to test whether altered retinal BDNF concentrations led to adaptive changes in BDNF TrkB receptor protein availability in the retina, we performed ELISA of TrkB in the same rats.

In the intact rats, BDNF concentrations in the retina samples were in the high femtomolar range, and amounted to 1.11–1.50 ng/mg protein (Left + Right mean = 1.29 ± 0.10; *n* = 9) (Figure 4A).

Six weeks after the induction of unilateral glaucoma, there was a bilateral decrease of BDNF concentration in the retina, in five of the six rats. The BDNF concentration reached, on average, 70% of control values (one-way ANOVA with Welch correction, *F* (4,28) = 14.71, *p* = 0.0001; Games–Howell post-hoc test; non-treated retina *p* = 0.003, glaucomatous retina *p* = 0.138, vs. intact rats) (Figure 4A). The BDNF concentrations in the Left and Right retinas were highly and positively correlated (Pearson correlation coefficient *r* = 0.86, *p* = 0.029), indicating that the glaucomatous eye is not isolated and draws the line of changes in the contralateral eye.

AAV2-BDNF injection resulted in a marked, bilateral upregulation of BDNF concentration in the retina (*n* = 6); on the injected side it was over two-fold higher than in the intact rats (Games–Howell post-hoc test: non-treated retina, *p* = 0.056; glaucomatous retina, *p* = 0.046) and three times higher than in the rats with glaucoma, BDNF-untreated (Games–Howell post-hoc test; non-treated retina, *p* = 0.016; glaucomatous retina, *p* = 0.019 (Figure 4A). Again, the BDNF concentration values in the Left and Right retinas tended to be positively correlated (Pearson correlation coefficient *r* = 0.76, *p* = 0.081), suggesting some BDNF transport from the injected to the contralateral eye, dependent on BDNF production in the transfected cells and on the neurotrophin content in their cellular milieu.

In the intact rats, the retinal TrkB protein concentrations were in the picomolar range, and amounted to 2.17–3.58 ng/mg prot. (L + R mean = 2.92 ± 0.56; *n* = 9). Figure 4B shows that after the induction of glaucoma, there was a bilateral increase in the TrkB protein concentration in the retinas of all rats (*n* = 6), which was over four-fold higher (12.38 ± 5.84 ng/mg prot.) on the glaucoma side, and two-fold higher (6.70 ± 1.87 ng/mg prot.) on the contralateral side, than in the intact rats (one- way ANOVA with Welch’s correction, *F* (4,28) = 8.19, *p* = 0.0018; Games–Howell post-hoc test: glaucomatous retina, *p* = 0.052; non-treated retina, *p* = 0.019; L/R retina *p* = 0.272). A weak correlation between the TrkB protein levels in the Left and Right retinas (Pearson correlation coefficient *r* = 0.30, *p* = 0.564) indicated that the adaptive processes, reflected by the increased TrkB synthesis/maintenance in the glaucomatous and contralateral eyes, do not develop in parallel.

AAV2-BDNF injection led to the molecular changes restoring normal TrkB concentrations in the retina (non-treated: 3.21 ± 0.47 ng/mg prot., glaucomatous: 3.11 ± 0.69 ng/mg prot.; Games–Howell posthoc test: N.S. vs. intact rats). The TrkB concentration values in the Left and Right retinas were highly correlated (Pearson correlation coefficient *r* = 0.86, *p* = 0.028) suggesting a common, strong response to conditions evoked by increased BDNF supply. We queried the dependence of TrkB expression on BDNF concentration, and examined the correlation between BDNF and TrkB levels in the AAV2-BDNF-untreated and -treated groups. The analysis revealed that within a subset of data from the AAV2-BDNF-treated group, in which BDNF exceeds control levels, the correlation is positive (glaucomatous retina: *r* = 0.94, *p* = 0.017). That result indicated that in conditions of moderate BDNF overexpression in the retina, the TrkB protein does not undergo downregulation, but follows BDNF levels, which may be of significant functional importance.

## 3. Discussion

We have shown for the first time that in the microbead trabecular occlusion model of glaucoma characterized by an initial severe glaucoma attack [30], a single AAV2-BDNF intraocular injection, when given 3 weeks prior to glaucoma induction, significantly protects RGCs from degeneration. That effect, detected both in the central and peripheral regions of the retina, was achieved with a moderate titer (2 μL; 10^13^ GC/mL) of AAV2-BDNF, causing BDNF protein overexpression, which was maintained at moderate levels for at least 9 weeks from transgene administration.

We decided to inject AAV-BDNF three weeks prior to glaucoma induction to ensure stable BDNF overexpression in the retina prior to glaucomatous changes development. This was based on the data of others [32] and on our own experience [33], showing that in different models the stable expression is achieved around 2 weeks after injection. This management strategy was due to the lack of detailed data on the dynamics of retinal ganglion cell mortality as a result of a sudden, radical increase in IOP. In the event that most cells die shortly after the IOP reaches its highest value, the administration of AAV-BDNF at that time or later would frustrate the effect of gradually increasing BDNF concentration on the survival of these cells. While RGCs were shown to respond to an ischemic insult with transitional increase in the amount of BDNF associated with the putative retinal ganglion cells, that response is limited to several hours [1], and such a reaction is not expected to protect RGC long term. A study of the therapeutic window in the microbead trabecular occlusion model of glaucoma would be beneficial as regards a further understanding of the processes initiated by the treatment, and for the planning of the therapy.

Our hypothesis, that if overexpressed BDNF shows significant neuroprotective effects then its cognate receptor TrkB protein expression is not suppressed, has been supported. Namely, in conditions of moderate BDNF overexpression, the retinal concentration of the TrkB protein returned from the upregulated to control levels, and neither of the examined subjects showed TrkB downregulation. This data and our correlation analysis indicate that in the retinal tissue, when BDNF is within a range of moderately elevated levels, the TrkB protein levels are maintained within the control range. This relation is similar to the one of retinal gene expression assessed one year after single AAV-BDNF intraocular injection, showing that when *bdnf* is moderately upregulated (2.5-fold), the expression of the *trkb* gene (coding for Ntrk2 receptor) is close to intact controls [34].

A recent study that validated the neuroprotective efficacy of combined construct AAV2 TrkB-2A-mBDNF in a mouse model of ONI and a rat model of chronic IOP did not provide evidence that BDNF overexpression depresses TrkB signaling long-term. However, the observation was made that neuroprotection in conditions of the co-expression of BDNF and TrkB was higher than in conditions of single BDNF or TrkB therapy [35]. Surprisingly, in their examination of TrkB receptor responses, the authors showed changes of the 92 kDa protein, which was close to described previously = TrkB-TK- truncated form, and they did not detect either the 143–145 kDa full length (FL) form or its deglycosylated forms [22], Figures 2 and 3 in [35]. Thus, the question concerning the TrkB-FL response to BDNF overexpression remains suspended. Our findings and those of the others [21,35] provide the impetus for undertaking studies to examine in detail the spatio-temporal TrkB responses, including TrkB’s binding properties and TrkB receptor phosphorylation dynamics, in glaucoma and after treatment with a range of doses of AAV-BDNF, to establish optimal conditions for potential use in clinical trials.

Importantly for a better understanding of the mechanisms underpinning the efficacy of BDNF signaling in the retina, we have also shown that elevated concentrations of the TrkB receptor protein, which undergoes significant upregulation in rats with glaucoma, are not sufficient for signaling towards the protection of RGCs. Our result, which points to the importance of other activation-related steps, i.e., receptor phosphorylation and the availability of down-stream signaling molecules, is in line with recent observations that overexpressing TrkB from AAV2-TrkB alone, prior to glaucoma induction, is not sufficient to significantly improve RGC survival [35].

It is worth noting that in our study, 15% to 20% of RGCs were protected in AAV2-BDNF-treated rats, indicating that not only the largest retinal cells (with somal areas >250 μm^2^; ca 5% of total RGC population) are preferentially responsive to elevations of BDNF, as suggested by Mey and Thanos [8], but also the other ones respond. That the largest RGCs are retained with low BDNF doses, and medium-sized RGCs require a higher BDNF dose, has been confirmed in cats [9], suggesting the differential sensitivities of various RGC subpopulations to BDNF.

Attention should be paid to the intriguing result of a bilateral decrease in BDNF concentration in retinas after unilateral glaucoma induction, and of the bilateral increase in BDNF after unilateral transgene administration. The BDNF levels in L and R retinas were strongly correlated, which suggests that dysfunction of the glaucomatous eye is not isolated, and draws the line of changes in the contralateral eye in which no glaucoma was induced. It has been known that unilateral optic nerve lesions induce diffuse responses that cause an inflammatory and microglial reaction in the contralateral uninjured retina (reviewed in Ramirez, et al. [36]). Only recently, unilateral axotomy in mice was shown to bilaterally downregulate RGC-specific genes, and to cause a significant loss of RGCs in the contralateral retinas [37]. What are the pathways to the contralateral changes? A direct response to the death of the retino–retinal RGCs is possibly a minor factor, because the number of retino–retinal connections in adult albino rats is very small (0–4). Stress signals released from the injured retino–retinal RGCs, that would activate an inflammatory neurotoxic response, as well as the propagation of the glial reaction to the contralateral retina through the optic chiasm, and finally a reaction to deafferentation of the ipsilateral superior colliculus, are other possibilities. The retina is supported by the fenestrated choroidal and the inner retinal vasculature. The endothelial cells of the latter provide a tight inner blood–retinal barrier (BRB), essential for the maintenance of normal retinal structure and function. The breakdown of the BRB, which is a pathological sign of several major vision-threatening vascular diseases, like diabetic retinopathy, age-related macular degeneration and retinopathy of prematurity [38], might modify the RGC microenvironment, and may contribute to a contralateral transfer of overproduced BDNF.

The microbead occlusion model of glaucoma, which proved its reproducibility in our hands and was recently used in a non-human primate with a visual system similar to our own, represents an attractive model for determining the impact of BDNF interventions on TrkB responses to glaucoma [28,30,39]. The value of the model is the possibility of scaling the pressure with microbeads and the injection speed; more short-lived models with IOP elevation in the 30% range, and more long-lived models with an abrupt, high elevation of IOP, were described [30]. In our study, at least one-fold IOP elevation after microbead single injection lasted for four weeks (10 mmHg vs. 20 mmHg at 4 weeks), but if the injection is repeated, a similar degree of increase IOP (20 mmHg vs. 38 mmHg) may be maintained for 9 months, allowing for long-term analysis of changes [39,40]. We showed that the IOP elevation in both Glaucoma and BDNF transgene-expressing groups followed a similar pattern, in line with observations made in mice subjected to optic nerve crush or laser-induced OHT [16,35]. Taken together, our results and those of the others suggest that BDNF signaling does not influence the mechanisms underpinning aqueous humor circulation in the eye, indicating the value of combinatory therapy adopting treatments designed to reduce IOP and neurotrophin supplementation.

While many studies have shown that glaucoma causes atrophy and the loss of RGCs, there are less data showing that neurodegenerative processes in glaucoma extend beyond the eyeball [41,42,43]. As the terminals of 90% of the axons of the retinal ganglion cells, which form the optic nerve, are located in the lateral geniculate nucleus (LGN), there is a strong anatomical basis for the transfer of dysfunction to this station of the visual pathway. Neuropathological examinations have shown that shrinkage and the loss of neurons also occur within LGN, affecting all its layers [41,42]. Important for the depth of the visual impairment, the transneuronal transfer of impaired function from damaged neurons to the healthy ones [43,44], resulting in a reduced size and a depletion of the dendritic tree within LGN, has been observed also in the visual cortex (VCtx) after monocular enucleation [45]. Damage to the RGCs therefore has a profound effect on distant but related neurons within the brain’s visual structures [46]. In primate glaucoma, an increased intraocular pressure with non-significant damage within the optic nerve induces neuronal shrinkage within LGN [42]. As a result, changes in the metabolic activity of the VCtx occur. A meta-analysis of over 1900 studies performed using non-invasive neuroimaging techniques in over 2300 people with glaucoma has shown that those patients experience neurodegenerative changes throughout the entire visual system [47]. These changes primarily involve the reduced thickness and volume of the cerebral cortex. Therefore, glaucoma should be considered as a disease of not only the eye, but of the entire visual pathway. Although in our study neither the BDNF nor the TrkB protein concentrations in rats with glaucoma were changed in VCtx (not shown), further spatio-temporal studies are needed to construct a more complete image of the neurotrophic alterations within the system [48,49].

## 4. Materials and Methods

### 4.1. Animals

The experiments were carried out on 39 7–8 week old male Wistar rats, weighting 250–300 g at the beginning of the experiment. The animals had been purchased from the Mossakowski Medical Research Center of Polish Academy of Sciences in Warsaw, Poland. They were housed under standard humidity and temperature conditions on a 12 h light/dark cycle at the Nencki Institute of Experimental Biology Animal Facility, with free access to water and food. Experimental protocols involving animals were approved by the First Local Ethics Committee in Warsaw (no 625/2018), in compliance with the guidelines of the Directive 2010/63/EU of 22 September 2010 on the protection of animals used for scientific purposes. Three animals treated with microbeads did not show increased IOP and were not included in the study. The animals were divided into groups: Naive Control (CN, *n* = 12), Glaucoma (unilaterally induced glaucoma, *n* = 14) and AAV-BDNF + Glaucoma (unilaterally induced glaucoma preceded by intraocular injection of AAV-BDNF transgene, *n* = 13).

### 4.2. Microbead Unilateral Glaucoma Induction

The glaucoma model was developed as described by the Lewin-Kowalik Group [30], who adapted the model first described by Urcola et al. [28]. For glaucoma induction, an intracameral injection of 2.5% polystyrene microbeads in phosphate buffered saline (PBS) was given. The microbeads mechanically block the trabecular meshwork in the iridocorneal angle, limit the free flow of aqueous humor and cause a high pressure in the anterior chamber (AC). The combination of microspheres with diameter (Ø) of 10.0 μm and 6 μm was chosen as the most efficient for obtaining a highly and constantly elevated IOP in rats [30] leading to transient corneal edema and RGCs loss.

### 4.3. Injection Procedure

On the day of surgery, baseline intraocular pressure (IOP) readings were taken. Next, the animals were anesthetized in a surgery room with an intraperitoneal injection of ketamine (50 mg/kg body weight, Biowet, Puławy, Poland) and medetomidine (0.25 mg/kg body weight). Topical anesthesia (proxymetacaine hydrochloride, Alcon) was applied to the corneal surface to reduce pain-related mydriasis, and improve the penetration of the beads into the iridocorneal angle.

A glass capillary was attached to a 5 μL Hamilton syringe using plastic tubing. The whole set (capillary, plastic tubing and syringe) was filled with low-viscosity microscope immersion oil. Next, through the capillary tip (Ø 80 μm) 5 μL of 1.6% viscoelastic solution, a 5 μL suspension of microspheres Ø 10.0 μm and a 5 μL suspension of microspheres Ø 6.0 μm was pulled. This order allowed us to inject first the smaller, then the larger beads, which ensured their better penetration into the irydocorneal angle and trabecular meshwork, and finish with the viscoelastic to close the corneal wound and limit the outflow of microbeads suspension.

The lower eyeball conjunctiva was grabbed with microforceps to immobilize the eyeball (Figure 5). A glass capillary (Figure 5A) was introduced into the anterior chamber near the corneal limbus (Figure 5B) at an angle of approximately 45 degrees to the corneal surface in the lower quadrant of the cornea.

The entire volume was injected rapidly, because bead distribution was better this way than with slow injection, and resulted in higher IOP levels and a deeper loss of RGCs. The needle was kept in the AC for 3 min after injection. The antibiotic ointment (ofloxacin, 0.3%, Bausch&Lomb Floxal, Orion Pharma Ltd., Warsaw, Poland) was applied onto the corneal surface.

### 4.4. Monitoring of the Intraocular Pressure

Measuring intraocular pressure was an essential step in monitoring the glaucoma. The measurements were performed without anesthesia with a tonometer designed for rodents (Icare, TonoLab Wet.; Colonial Medical Supply, Espoo, Finland), whereby a light probe is used to rebound against the corneal surface to make fleeting contact. The resilience of the rebound is processed by tonometer software into the IOP value. IOP was measured binocularly before the glaucoma, on the day of glaucoma induction, after 3 and 7 days, and then once a week until the animals were culled. The software allowed for the elimination of the outliers and the presentation of the average value from five measurements.

### 4.5. AAV Vector Injection

The AAV2-BDNF vectors containing Woodchuck Hepatitis Virus Posttranscriptional Regulatory Element (pAAV2-CMV>hBDNF [NM_001143810.1]: WPRE) were produced and supplied by VectorBuilder Inc. (Chicago, IL, US). Rats were anesthetized with ketamine and medetomidine and given topical Alcon eye drops as described for microbead injection. The AAV2-BDNF vector (2 μL of prep. 10^13^ GC/mL) was injected intravitreally at the pars plana spot 3 weeks prior to glaucoma induction. All injections were carried out by the same surgeon.

### 4.6. Tissue Processing for Immunofluorescence

After 6 weeks from glaucoma induction the rats were deeply anesthetized with a lethal dose of Morbital (pentobarbital; 80 mg/kg body weight, intraperitoneal, Biowet, Puławy, Poland) and received transcardial perfusion with 400 mL of ice-cold 0.1 M PBS with heparine (0.4 mL), followed by 150 mL ice-cold paraformaldehyde solution (4% PFA in 0.1M PBS pH 7.4). The eyeballs were removed with retro-bulbar stumps of optic nerves, and post-fixed in ice-cold 4% PFA for several hours. After triple PBS rinsing, the retinas were isolated and incised to facilitate their flat arrangement on the glass slides. The retinas were stored in PBS in 4 °C for 24 h.

### 4.7. Immunofluorescence

For the quenching of tissue autofluorescence, the retinas were incubated in 100 mM glycine solution (45 min shaking, RT). After PBS washing (3×) the retinas were incubated with 20% normal goat serum (NGS) in PBST for 45 min to eliminate non-specific binding. Subsequently, the retinas were incubated with a monoclonal primary antibody recognizing BRN3a antigen (sc-8429, 1:200; Santa Cruz Biotechnology, Dallas, Texas, US), in 4% NGS in PBST, for 24 h. The Brn3a protein is a strong nuclear marker for the rodent RGCs population; in the rat this antibody labels repetitively 96% of the RGCs population [50,51,52]. Next, the retinas were washed with PBS (3×) and incubated with secondary anti-mouse antibody conjugated with Alexa Fluor 555 (A32732, 1:300; ThermoFisher Scientific, Waldham, MA, US) in 4% NGS in PBST (2 h shaking at RT). To visualize cell nuclei, the retinas were incubated for 15 min with DAPI (2 mg/100 mL, Sigma, St. Louis, MO, US). After extensive washing with PBS (6 h shaking, 4 °C) the retinas were transferred to glass slides, mounted with Vectashield antifade mounting medium (Vector Laboratories, Burlingame, CA, US) and kept in the dark at 4 °C until analysis.

### 4.8. Image Acquisition and RGC Counting

Images of retinas were captured with a Zeiss Spinning Disc confocal microscope (Carl Zeiss, Jena, Germany), using a 10× (for the whole retina image reconstruction) and 20× (for selected fields) objective lens. Z-stacks of 16-bit images consisted of 4–10 optical slices collected at 1 μm intervals and processed with a Max-projection tool.

The quantification of the immunolabeled RGCs was performed using ImageJ open source image processing software in four quadrants of flattened retinas: in the peripheral regions located about 2.5 mm from the optic disc, and the central parts, centered at about 1 mm diameter away from the center of the optic disc (Figure 2). Because Brn3a marks predominantly cell nuclei, Brn3a immunolabeled objects with an area between 20 and 200 μm², covering the whole range of the RGC soma’s diameter [53], were counted. All cells detectable in the region of interest were taken into account. Cells were counted in 300 μm × 300 μm squares placed in regions with no signs of discontinuity or artifacts created during processing (16–64 fields; total area 1.44–5.76 mm^2^), and their density was calculated per 1 mm² area.

### 4.9. Tissue Dissection for ELISA

The rats were deeply anesthetized with a lethal dose of pentobarbital and quickly perfused transcardially with 400 mL of ice-cold PBS. The brain and eyeballs were removed from the skull and placed on ice. Both the left and right visual cortex and the retinas were isolated and frozen on dry ice. Dissected tissues were stored at −80 °C.

### 4.10. Preparation of Homogenates and BDNF and TrkB ELISA Assay

A ChemiKineTM BDNF Sandwich ELISA Kit (CYT306, Millipore, Burlington, MA, US) detecting both human and rat BDNF protein was used, according to the manufacturer’s instructions. Crude tissue homogenates (20% w/v) were prepared in 100 mM Tris buffer (pH 7.0), containing 2% BSA, 1M NaCl, 2% Triton X-100 and Complete Protease Inhibitor Cocktail, 200 μm phenylmethyl-sulphonyl fluoride (PMSF; Sigma-Aldrich) and 157 μg/mL benzamidine hydrochloride (Serva, Westmont, IL, US). An IKA Ultraturrax (Staufen, Germany) tissue grinder was used to disrupt the tissue. The homogenates were incubated on ice for 30–60 min and centrifuged at 15,000× *g* for 30 min at 4 °C. ELISA was performed on supernates diluted 50×. All samples were run in duplicate.

For TrkB ELISA (LS-F24219, Rat NTRK2/TRKB Sandwich ELISA Kit, LifeSpan Biosciences Inc., Seattle, WA, US), tissue homogenates prepared for BDNF ELISA and stored at −18 °C were thawed, re-homogenized and centrifuged at 2000× *g* for 10 min at 4 °C. ELISA was performed on supernates diluted 10× following the manufacturer’s instructions. All samples were tested in duplicate.

### 4.11. Protein Assay

Protein concentration was determined by the modified Bradford method in cuvettes [54] with bovine serum albumin (BSA) as a standard. Briefly, 0.1 mL of BSA solution containing 1–10 pg protein was pipetted into tubes in triplicates. One mL of Bradford reagent was added to each test tube. Absorbance at 595 nm was measured in a Biospectrometer (Eppendorf, Hamburg, Germany). The supernates of samples were diluted 200× with dH_2_0, sampled and measured as above. A standard curve was prepared and a regression coefficient was calculated to determine the linearity of the curve. The absorbances of the experimental samples were compared with the values from the standard curve, and the total protein concentration in each sample was calculated.

### 4.12. Statistical Analysis

The Shapiro–Wilk test was used to verify the normality of data distribution. The homogeneity of variance in the groups was analyzed with the Leven and Bartlet’s tests. Two-way ANOVA with repeated measurements followed by a Sidak’s multiple comparisons test was used to compare the density of RCG between groups. Because the homogeneity of variance was violated in sets of data on BDNF and TrkB levels in some groups, one-way ANOVA with Welch’s correction followed by Games–Howell pos-hoc tests were used. For the comparison of IOP levels in time, a *t*-test for dependent variables was used. STATISTICA 13.1 (StatSoft. Inc. Tulsa, OK, USA) and R software were used for the data analysis.

## Figures and Tables

**Figure 1 ijms-21-06262-f001:**
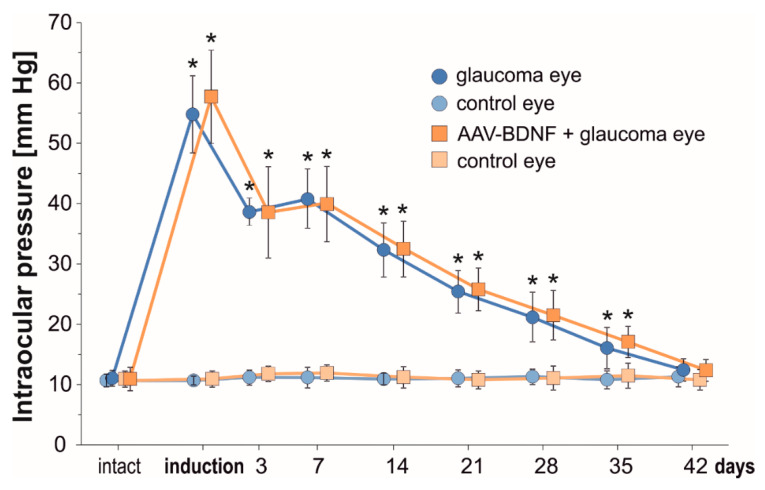
The temporal course of intraocular pressure (IOP) from the first till 42 days after uniocular (Right eye) microbead injection. In the Right eye there is a significant increase in the IOP immediately after glaucoma induction. In the Left untreated eye, IOP remains unchanged and is maintained within a range of pre-injection values. AAV2-BDNF administration to the Right eye preceded glaucoma induction by 3 weeks. Data are expressed as mean ± SD from 8 (glaucoma group) and 7 (AAV2-BDNF+ glaucoma group) rats. * *p* < 0.01, different from intact and between time-points (except for day 3 vs. 7). For the comparison of IOP levels in time, a *t*-test for dependent variables was used.

**Figure 2 ijms-21-06262-f002:**
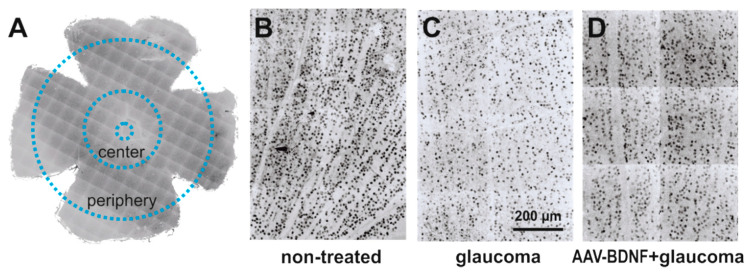
The effect of glaucoma and AAV-BDNF treatment on the retinal ganglion cell (RGC) number was examined by counting RGCs in 300 μm × 300 μm m fields in the central and peripheral parts of the retina (**A**). Images from non-treated eye (**B**) were compared to images from contralateral glaucomatous (**C**) or glaucomatous and AAV-BDNF treated eyes (**D**). RGCs were immunolabeled with anti-Brn3a antibody, a widely accepted marker of rodent RGC [31].

**Figure 3 ijms-21-06262-f003:**
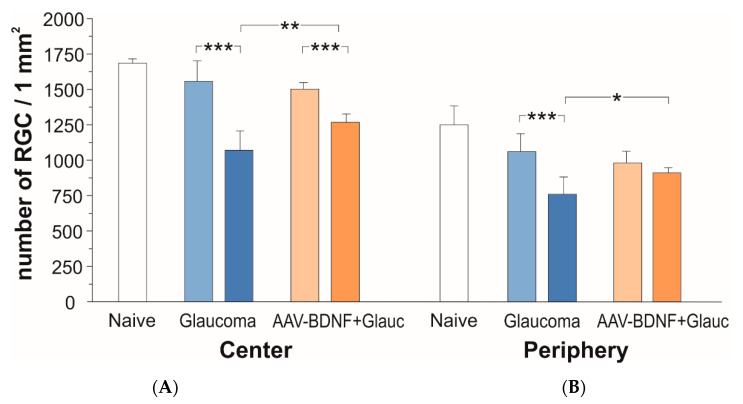
The effect of glaucoma (blue) and AAV2-BDNF treatment (orange) on the number of RGCs in central (**A**) and peripheral (**B**) fields of the retina six weeks after glaucoma induction. Dark color bars refer to the Right, glaucomatous eye. Light color bars mark the Left fellow eye. Data are shown as mean ± SD from 8 (glaucoma group) and 7 (AAV2-BDNF+ glaucoma group) rats. Two-way RM ANOVA, Sidak’s post-hoc test; *** *p* < 0.001, ** *p* < 0.01, * *p* < 0.05.

**Figure 4 ijms-21-06262-f004:**
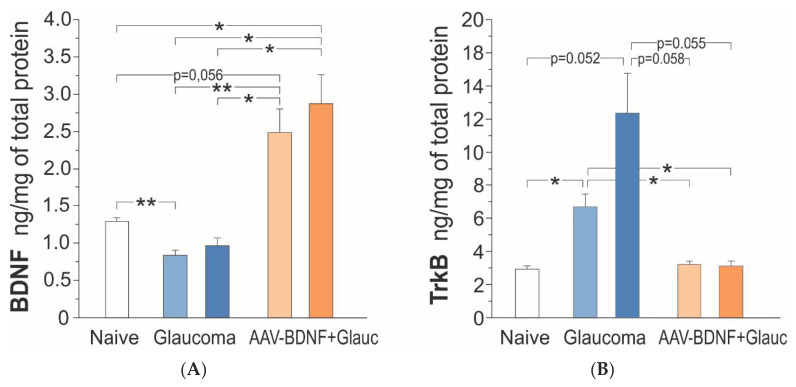
BDNF (**A**) and TrkB (**B**) protein concentration in the retina of the rats: intact control (Naive; white), with uniocular, transient glaucoma (Glaucoma; blue) and injected intravitreal with AAV-BDNF three weeks prior to glaucoma induction (AAV-BDNF + Glauc.; orange). Light color represents the Left eye, dark color represents the Right eye with induced glaucoma. Data are shown as mean ± SD from 9 (intact control group), 6 (glaucoma group) and 6 (AAV2-BDNF + glaucoma group) rats. ** *p* < 0.01, * *p* < 0.05.

**Figure 5 ijms-21-06262-f005:**
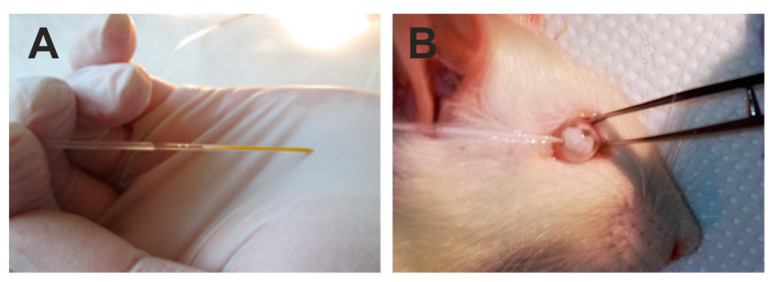
Illustration of the steps of the surgery of intraocular introduction of polystyrene microbeads. Injection was performed with a glass capillary filled with a suspension of viscoelastic and polystyrene microbeads (**A**) into the anterior chamber of the rat eye (**B**).

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
