# Peer review of "Neuroprotection of Retinal Ganglion Cells with AAV2-BDNF Pretreatment Restoring Normal TrkB Receptor Protein Levels in Glaucoma"

_ijms, 2020, doi:10.3390/ijms21176262_

Round 1

Reviewer 1 Report

The present study explores the interaction between BDNF and TrkB levels in relation to the intraocular pressure. The authors found that unilateral glaucoma leads to bilateral decrease of BDNF concentration accompanied by up to TrkB regulation. Moreover they found that long-term moderated BDNF overexpression in glaucomatous eye restores normal TrkB levels driving RGC neuroprotection.

The main concern to the present paper is that the administration of AAV2-BDNF take place 3 weeks before inducing the experimental glaucoma, and it is very possible that during this very long period of time the AAV2 will go travel through the bloodstream reaching the contralateral eye. Authors do not discuss why did the treatment before the damage and if they have experience the injection after the damage what were the results. It is very difficult to think about a possible treatment when the agent has to be apply before the damage. There are previous studies in which the changes in BDNF have been quantify after the damage, so the interesting point will be to study that therapeutic window.

In general the article is interesting even not relevant for future treatments as explained previously. The manuscript lack of precision in the bibliographic citations and that has to be corrected.

Introduction: Previous studies by the group of Vecino et al. on BDNF and RGCs should be cited. It is important to indicate in the introduction that BDNF and TrkB are co-localized in the cell body of the RGCs (Vision Res. (2002) 42(2) 151-17) to understand the purpose of the study. Several articles have pointed out the increase of BDNF expression by RGCs after insults like NMDA injection (Neuroreport(1999) 10(5)1103-6) or by an ischemic insult (Gen. Pharmacol. (1998) 3:305-314). In this last case, the increase of BDNF is transient and it will be interesting to discuss in the present study how if BDNF is artificially increased 3 weeks before the damage how this increase could be the reason of the neuroprotection.

In Material and Methods section, the authors should cite the first description of the microbead trabecular occlusion model of glaucoma that was first published by Urcola et al. Exp. Eye. Res. 2006. The authors should change the cite (26) in lines 68, 98, 201, 265, 307 by the (28). The modified posterior version of the method published by the authors, in cite 26, is not significant different method, is an adaptation or modification. The authors for that reason have to cite the original method as it is done with the Bradford method in line 407, in which is mentioned that the method used in the present study is a modified version and then, cite the original method (49).

The number of animals used for each technique should be mentioned. How many animals were used for the quantification of the RGCs after treatment with AAV and how many for ELISA and also if only the animals that behave the same in terms of IOP response, were selected for the study.

Author Response

Responses to Reviewer 1 comments:

  1. The main concern to the present paper is that the administration of AAV2-BDNF take place 3 weeks before inducing the experimental glaucoma, and it is very possible that during this very long period of time the AAV2 will go travel through the bloodstream reaching the contralateral eye.
  2. Authors do not discuss why did the treatment before the damage and if they have experience the injection after the damage what were the results. It is very difficult to think about a possible treatment when the agent has to be apply before the damage. There are previous studies in which the changes in BDNF have been quantify after the damage, so the interesting point will be to study that therapeutic window.

RESPONSE to 1 & 2:

We are aware that the addressed issues might be of importance for the physiology of the eye; it was taken into consideration when the experiment was planned.

According to the available data, the half-life of the AAV viruses ranges from minutes to several hours. In particular, AAV2/1, AAV2/2 AAV2/5 and AAV/8 are maintained in the blood from of minutes till 11 hours and after systemic injection of 1x109genomic  copies full clearance within 2 days was reported (van Gestel et al.,  PLoS One. 2014; 9(5): e97639).

However, we admit that there might be some spread of BDNF, when overexpression takes place. That possibility has been discussed in the manuscript (lines 257-260) with reference to the glaucomatous state. Before glaucoma is initiated, the possibility is minor. That is because in  the blood-ocular barrier system, which  is formed by the blood-aqueous barrier and the blood-retina barrier (BRB), the latter is particularly tight and restrictive, forming a physiological barrier that regulates ion, protein, and water flux into and out of the retina. BRB also becomes a reason that systemic administration of some drugs is not suitable for the treatment of retinal diseases, and that intraocular drugs act selectively in the eyeball.

Because we had to strike the right balance between (1) the period when the stable expression of the transgene is achieved and (2) the time of microbead injection which induces neurodegeneration and might affect AAV-BDNF spread, leakage and RGCs infection, we decided to  optimize conditions and introduce a transgene 3 weeks prior to glaucoma induction. That was based on the data of the others (Fisher et al., 1996) and on our experience (Platek et al., 2019) showing that in different models the stable expression is achieved around 2 weeks after injection. That information has been added to the DISCUSSION CHAPTER (lines 210-217) in the revised version of the manuscript. We agree that the study of the therapeutic window would be beneficial for further understanding of the processes turned on by the treatment. That comment has been added as well (l. 217-219).

  1. The manuscript lack of precision in the bibliographic citations and that has to be corrected.

A) Introduction: Previous studies by the group of Vecino et al. on BDNF and RGCs should be cited. It is important to indicate in the introduction that BDNF and TrkB are co-localized in the cell body of the RGCs (Vision Res. (2002) 42(2) 151-17) to understand the purpose of the study.

RESPONSE:

Thank you for that remark. We fully agree with the importance of that study and its relevance to our work. We added the citation of Vecino et al. paper accordingly (no 1 in the revised manuscript) and introduced a sentence on BDNF and TrkB expression in RGC, which provides the basis for the autocrine/paracrine BDNF signaling.

B) Several articles have pointed out the increase of BDNF expression by RGCs after insults like NMDA injection (Neuroreport(1999) 10(5)1103-6) or by an ischemic insult (Gen. Pharmacol. (1998) 3:305-314). In this last case, the increase of BDNF is transient and it will be interesting to discuss in the present study how if BDNF is artificially increased 3 weeks before the damage how this increase could be the reason of the neuroprotection.

RESPONSE:

Indeed, the transient increase of BDNF (and TrkB) is worth consideration in the context of long-term neuroprotection. We added a short paragraph on that to the DISCUSSION CHAPTER, not to speculate much on that early period having no data on it.

In Material and Methods section, the authors should cite the first description of the microbead trabecular occlusion model of glaucoma that was first published by Urcola et al. Exp. Eye. Res. 2006. The authors should change the cite (26) in lines 68, 98, 201, 265, 307 by the (28). The modified posterior version of the method published by the authors, in cite 26, is not significant different method, is an adaptation or modification. The authors for that reason have to cite the original method as it is done with the Bradford method in line 407, in which is mentioned that the method used in the present study is a modified version and then, cite the original method (49).

RESPONSE:

We agree with the comment. The respective changes have been introduced to the text.

  1. The number of animals used for each technique should be mentioned. How many animals were used for the quantification of the RGCs after treatment with AAV and how many for ELISA and also if only the animals that behave the same in terms of IOP response, were selected for the study.

RESPONSE:

Out of 30 animals treated with microbeads to induce glaucoma 3 rats did not show increased IOP and these were eliminated from further analysis. As presented in the Material & Methods chapter, there were three animal groups: Naive Control (CN, n = 13), Glaucoma (unilaterally induced glaucoma, n = 14), and AAV-BDNF + Glaucoma (unilaterally induced glaucoma preceded by intraocular injection of AAV-BDNF transgene, n = 13) (lines 317-319). To clarify the information on the number of animals used for quantification of the RGCs and for ELISA, we:

a) Modified the description in the Figure 1 legend, matching it with the description in the Figure 2 legend (there were 8 rats in the glaucoma group and 7 rats in the AAV2-BDNF + glaucoma group).

b) Added the numbers to the Figure 4 legend (there were 6 rats in glaucoma group and 6 rats in the AAV2-BDNF + glaucoma group, plus 9 rats in the control group).

c) Removed 4 rats from the control group, which were used to optimize staining and conditions of ELISA.

Moreover, we checked once again and corrected grammar and punctuation mistakes in the manuscript.

Reviewer 2 Report

This is a well written and organized manuscript; however, I have few concerns that require the attention of the authors. The authors should ensure that sentences with minor grammatical/punctuation errors are revised. Ensure that all abbreviations or acronyms used in the manuscript for the first time are written in full. Concerning reference to TrkB in the manuscript, specify whether you are referring to TrkB protein or TrkB receptor. I would like the author(s) to address these suggestions/comments raised by the reviewer.

Specific suggestions/comments raised by the reviewer:

Line 71: Elaborate on this sentence. Do you intend to say 'we show that unilateral glaucoma... or We will show that unilateral ..."

Lines 76 - 79: The authors should reword this sentence in order to render it more comprehensible.

In light of the statement made in lines 194 – 196, elaborate on the reason for the lack of bilateral decrease of BDNF concentration in the retina observed in one of the rats (see line 164). As stated by the authors in line 180 “there was a bilateral increase of TrkB protein concentration in retinas of all rat”

Line 166: There is inconsistent labelling of figures. The authors should indicate which figure is Figure 4A and Figure 4B. Is there a figure 5? Please specify.

Lines 175 - 176: Elaborate on the propensity of a positive correlation of BDNF concentration values in the Left and Right retinas in the light of a Pearson correlation coefficient r = 0.76 and p = 0.081.

Lines 192 - 194: reword the sentence.

Lines 219 - 222: The authors should reword this sentence in order to render it more comprehensible.

Author Response

Responses to REVIEWER 2 comments

Line 71: Elaborate on this sentence. Do you intend to say 'we show that unilateral glaucoma... or We will show that unilateral ..."

RESPONSE:

The sentence has been changed as requested.

Lines 76 - 79: The authors should reword this sentence in order to render it more comprehensible.

RESPONSE:

The sentence has been divided into two statements to make them more comprehensible.

In light of the statement made in lines 194 – 196, elaborate on the reason for the lack of bilateral decrease of BDNF concentration in the retina observed in one of the rats (see line 164). As stated by the authors in line 180 “there was a bilateral increase of TrkB protein concentration in retinas of all rat”

RESPONSE:

Indeed, we do not know the reason for the lack of bilateral decrease of BDNF concentration in the retina observed in the rat no. 4.1. TrkB concentration in that rat was bilaterally increased and was the highest in the glaucomatous eye in the group. We were considering the possibility that in that rat the glial response developed leading to increased synthesis of both proteins. There is a paper (Graefes Arch Clin Exp Ophthalmol 2016) showing increased BDNF and trkB in the Muller and bipolar cells after ischemia. However we had nomeans to prove it as the material was taken for biochemical assays.

Line 166: There is inconsistent labelling of figures. The authors should indicate which figure is Figure 4A and Figure 4B. Is there a figure 5? Please specify.

RESPONSE:

The mistake in the line 166 (now 172) has been corrected. The labeling has been corrected accordingly and the description in the Figure 4 legend has been improved (lines 161-168 of the revised manuscript).

Figure 5 is an illustration of the intraocular injection of microbeads and belongs to the Material and method section, so it is placed there (lines 351-355 of the revised manuscript).

Lines 175 - 176: Elaborate on the propensity of a positive correlation of BDNF concentration values in the Left and Right retinas in the light of a Pearson correlation coefficient r = 0.76 and p = 0.081.

RESPONSE:

Our proposition of processes and possible pathways which led to bilateral upregulation of BDNF concentration, has been elaborated in the DISCUSSION section (l. 272-278 of the revised manuscript). As proposed, BDNF transport from the injected to the contralateral eye is dependent on BDNF production in the transfected cells and its release, but also on the blood-retinal barrier changes on the way. Therefore BDNF level in the contralateral eye is not a simple function of BDNF level in the glaucomatous eye what might explain the probability of correlation, which fell short of the required level of significance.  

We would like to comment on that whether AAV2 itself would be transported in the bloodstream during weeks from the injection and infect contralateral eye. That possibility is minor in light of data on AAV2/1, AAV2/5 and AAV/8 which showed that half-life of these vectors in the blood ranges from tens of minutes till 11 hours; after systemic injection of 1x109 genomic copies full clearance within 2 days was reported (van Gestel et al.,  PLoS One. 2014; 9(5): e97639). Another limiting factor is blood-retinal barrier which prior to microbead injection is intact. Therefore at the day of microbead injection and afterwards the only source of transgenic protein are the cells which produce the transgene.

Lines 192 - 194: reword the sentence.

RESPONSE:

The sentence has been reworded (lines 198-200 in the revised manuscript).

Lines 219 - 222: The authors should reword this sentence in order to render it more comprehensible.

RESPONSE:

The sentence has been divided into two statements and reworded (lines 238-241 in the revised manuscript).

Moreover, we added corrected grammar and punctuation mistakes in the manuscript.